

# Genetic connectivity between Atlantic bluefin tuna larvae spawned in the Gulf of Mexico and in the Mediterranean Sea

Carolina Johnstone[1], Montse Pérez[2], Estrella Malca[3,4], José María Quintanilla[1], Trika Gerard[4], Diego Lozano-Peral[5], Francisco Alemany[6], John Lamkin[4], Alberto García[1] and Raúl Laiz-Carrión[1]

[1] Centro Oceanográfico de Málaga, Instituto Español de Oceanografía, Consejo Superior de Investigaciones Científicas, Fuengirola, Málaga, Spain
[2] Centro Oceanográfico de Vigo, Instituto Español de Oceanografía, Consejo Superior de Investigaciones Científicas, Vigo, Pontevedra, Spain
[3] Cooperative Institute for Marine and Atmospheric Studies, University of Miami, Miami, Florida, United States of America
[4] Southeast Fisheries Science Center, National Marine Fisheries Service, National Oceanic and Atmospheric Administration, Miami, Florida, United States of America
[5] Centro de Supercomputación y Bioinnovación, Servicios Centrales de Apoyo a la Investigación, Universidad de Málaga, Málaga, Spain
[6] International Commision for the Conservation of Atlantic Tunas, Madrid, Spain

Corresponding author
Carolina Johnstone,
carolina.johnstone@ieo.es

## ABSTRACT

The highly migratory Atlantic bluefin tuna (ABFT) is currently managed as two distinct stocks, in accordance with natal homing behavior and population structuring despite the absence of barriers to gene flow. Larval fish are valuable biological material for tuna molecular ecology. However, they have hardly been used to decipher the ABFT population structure, although providing the genetic signal from successful breeders. For the first time, cooperative field collection of tuna larvae during 2014 in the main spawning area for each stock, the Gulf of Mexico (GOM) and the Mediterranean Sea (MED), enabled us to assess the ABFT genetic structure in a precise temporal and spatial frame exclusively through larvae. Partitioning of genetic diversity at nuclear microsatellite loci and in the mitochondrial control region in larvae spawned contemporarily resulted in low significant fixation indices supporting connectivity between spawners in the main reproduction area for each population. No structuring was detected within the GOM after segregating nuclear diversity in larvae spawned in two hydrographically distinct regions, the eastern GOM (eGOM) and the western GOM (wGOM), with the larvae from eGOM being more similar to those collected in the MED than the larvae from wGOM. We performed clustering of genetically characterized ABFT larvae through Bayesian analysis and by Discriminant Analysis of Principal Components (DAPC) supporting the existence of favorable areas for mixing of ABFT spawners from Western and Eastern stocks, leading to gene flow and apparent connectivity between weakly structured populations. Our findings suggest that the eastern GOM is more prone for the mixing of breeders from the two ABFT populations. Conservation of this valuable resource exploited for centuries calls for intensification of tuna ichthyoplankton research and standardization of genetic tools for monitoring population dynamics.

## INTRODUCTION

Bluefin tuna are epipelagic oceanic species that have been exploited globally for centuries (*Muhling et al., 2017*). Regional fisheries organizations manage Pacific bluefin *Thunnus orientalis* (Temminck and Schlegel, 1844) as one stock, Southern bluefin *T. maccoyii* (Castelnau, 1872) also as one stock, and Atlantic bluefin tuna (ABFT) *T. thynnus* (Linnaeus, 1758) as two, Western and Eastern, stocks. Each of these stocks is considered a distinct population (*Kumar & Kocour, 2015*). Although individual juvenile and adult fish are capable of trans-Atlantic migrations and share feeding grounds in the North Atlantic Ocean, the Western ABFT primarily spawn in the Gulf of Mexico (GOM) and the Eastern ABFT in the Mediterranean Sea (MED) (*Muhling et al., 2017*). Despite additional spawning areas for ABFT have been recently discovered in the Slope Sea and the Bay of Biscay (*Richardson et al., 2016*; *Rodriguez, Johnstone & Lozano-Peral, 2021*), the GOM and the MED are still considered the main grounds to which the adults return to breed, with regional currents linking larval and juvenile nursery habitats (*Muhling et al., 2017*).

Overexploitation of this valuable fishery resource resulted in sharp decreases in abundance from the 1960s onward. This led to the implementation of strict management measures after 2007 by the International Commission for the Atlantic Tuna (ICCAT), Madrid, Spain. The 2017 assessment results from the virtual population analysis (VPA) indicated that the spawning stock biomass (SSB) exhibited a substantial increase from the late 2000s (*ICCAT, 2020*), which has allowed to presently raise catch quotas, in spite the species is still currently listed as endangered by the International Union for Conservation of Nature (*IUCN, 2020*). Increasing our understanding of the connectivity between ABFT populations is crucial for conservation of ABFT. The existence of weak structuring between Western and Eastern stocks is supported by numerous studies that have assessed the population structure of ABFT through different approaches such as analysis of genetic diversity with multiple applications for conservation and fisheries management (*Abdul-Muneer, 2014*; *Ovenden et al., 2015*; *Cuéllar-Pinzón et al., 2016*). In particular, a variety of genetic markers have been employed to assess ABFT population dynamics including the fast-evolving maternally inherited mitochondrial DNA (mtDNA) (*Carlsson et al., 2004*, *2007*; *Alvarado Bremer et al., 2005*; *Boustany, Reeb & Block, 2008*), codominant highly polymorphic microsatellite loci (*Carlsson et al., 2004*, *2007*; *Riccioni et al., 2010*, *2013*; *Antoniou et al., 2017*), and more recently high-throughput sequencing of highly abundant single nucleotide polymorphisms or SNPs (*Antoniou et al., 2017*; *Puncher et al., 2018*; *Rodríguez-Ezpeleta et al., 2019*). Subtle structuring of ABFT populations across the Atlantic Ocean is the general conclusion of genetic studies, with significant $F_{ST}$ fixation indices that estimate partitioning of genetic diversity on a zero to one scale ranging from 0.005 to 0.012 (*Puncher et al., 2018*). The analysis of the phylogenetic signal of

mtDNA suggests a dramatic reduction in ABFT population size during Pleistocene glaciations, followed by sudden population expansion with an increase in gene flow to levels resulting in homogenization of stocks (*Alvarado Bremer et al., 2005*). Other disciplines besides genetics have confirmed that the two ABFT populations are indeed highly mixed. Estimation of natal origin through otolith microchemistry demonstrated substantial intermingling of individuals from both populations in North Western Atlantic waters (*Rooker et al., 2003*, *2008a*); spatio-temporal distributions of electronically tagged ABFT revealed overlapping of foraging grounds and provided evidence for spawning fidelity to the MED (*Block et al., 2005*), and a fisheries model predicts that mixing depends on season and location (*Taylor et al., 2011*). A recently developed population assignment method combining genetic (SNPs) and environmental (otolith microchemistry) markers revealed complexity in ABFT structure (*Brophy et al., 2020*). There is thus mounting evidence supporting the highly-mixed nature of ABFT populations. Nevertheless, certain questions still remain unresolved, including the level of gene flow or connectivity between populations and the exchange rate between stocks.

Larval cooperative studies have provided insights into multiple aspects of tuna larval ecology and biology, particularly growth and food web dynamics (*Laiz-Carrión et al., 2015*; *Malca et al., 2017*; *Laiz-Carrión et al., 2019*). Intensification of cooperative ichthyoplankton prospection surveys to collect ABFT larvae is currently required after the incorporation of larval indexes into stock assessment (*Ingram et al., 2017*), providing a valuable opportunity to gain further knowledge on other aspects such as tuna population dynamics. As far as we are aware, ABFT population genetic structuring has not yet been assessed exclusively from larval ABFT ensuring both correct geographical assignment and representation of genetic features of successful breeders. The study of ABFT population genetics has mostly focused on adults collected in Eastern and Western stocks (*Alvarado Bremer et al., 2005*) or within the Mediterranean Sea (*Riccioni et al., 2010*, *2013*; *Viñas et al., 2011*; *Vella et al., 2016*; *Antoniou et al., 2017*). Juveniles (young-of-the-year, YOY) collected in nursery areas have occasionally been characterized exclusively or together with adults (*Boustany, Reeb & Block, 2008*), and with larvae in only a few interesting studies (*Carlsson et al., 2004*, *2007*; *Puncher et al., 2018*; *Rodríguez-Ezpeleta et al., 2019*). The use of YOY juveniles collected in nursery habitats reduces the risk of including migrants in the incorrect subpopulation. However, these juveniles are strong swimmers and can move thousands of kilometres away from their spawning location. Conveniently, the collection of larvae in spawning grounds completely eliminates this potential error and provides the genetic signal of spawners. The movement of larval ABFT (<20 days age) is physiologically limited by multiple factors, including a lack of morphological development of fin complements and muscle required for locomotion outside of the corresponding spawning grounds. Full sibling removal is, however, recommended to accurately estimate genetic diversity, at least for larvae of other animals (*Goldberg & Waits, 2010*). The aim of our study was to assess ABFT genetic structure in a precise temporal and spatial frame exclusively through larvae collected in the two main spawning grounds.

Management and conservation of ABFT requires continuous research concerning the population dynamics and exchange between Western and Eastern stocks that can be achieved through tagging studies (*Rooker et al., 2019*) or genetic tools for traceability (*Puncher et al., 2018*; *Rodríguez-Ezpeleta et al., 2019*). Genetic structure between ABFT collected in Western and Eastern stocks is herein assessed by analyzing diversity in larvae through two differently inherited markers, mtDNA control region sequences and nuclear microsatellite loci, following the approach of *Carlsson et al. (2007)*. Fast evolving mtDNA is maternally inherited and provides the evolutionary signal for this highly migratory species, for which past hybridization events with albacore have been reported (*Alvarado Bremer et al., 2005*). Highly polymorphic nuclear microsatellite loci with Mendelian inheritance allow evaluating population connectivity. We analyzed larvae collected only during the 2014 spawning season to capture a snapshot of gene flow between ABFT breeders in the main spawning ground of each stock without blurring the image by mixing years with different stock mixing rates. Individual-based clustering analysis of larval ABFT genetic diversity indicates apparent connectivity between the GOM and MED spawning grounds that could support the hypothesis of mixing of breeders belonging to different stocks.

## MATERIALS & METHODS

### Field collection of ABFT larvae

We conducted larval ABFT collections during the peak of the reproductive season in the two main spawning areas for ABFT, the GOM and the MED (Fig. 1). The GOM was divided into subregions separated at 90°W meridian and designated as eastern (eGOM) and western (wGOM) (*Muller-Karger et al., 2015*). A total of 76 stations were sampled in the GOM cruise "WS1405" (approved by the National Oceanic and Atmospheric Administration, Washington, D.C., USA with Permit Number TUNA-SRP-14-02) carried out from 28 April to 20 May 2014 aboard the R/V F.G. Walton Smith. The "BLUEFIN14" cruise (approved by the Spanish Institute of Oceanography, Madrid, Spain (Instituto Español de Oceanografía) "BLUEFIN TUNA" project) explored a total of 123 stations and took place from 13 June to 3 July 2014 in the western MED on board the R/V SOCIB, of the Balearic Islands Coastal Observing and Forecasting System. Both cruises used similar standardized methodologies for field collection of fish larvae as described previously (*Laiz-Carrión et al., 2015*). Larvae were either frozen in liquid nitrogen or preserved in 96% ethanol immediately upon retrieval.

Larvae were identified as ABFT following morphological, meristic and pigmentation characters. Standard length (SL) was measured to the nearest 0.01 mm using Image J 1.44a (National Institute of Health, Stapleton, NY, USA). Developmental stage was determined following *Richards (2005)*. ABFT-positive stations were first determined for each survey (31 stations in the GOM and 63 in the MED), and a subset of stations was selected for genetic analysis from each spawning ground (15 stations in the GOM and 14 in the MED, see Fig. 1).

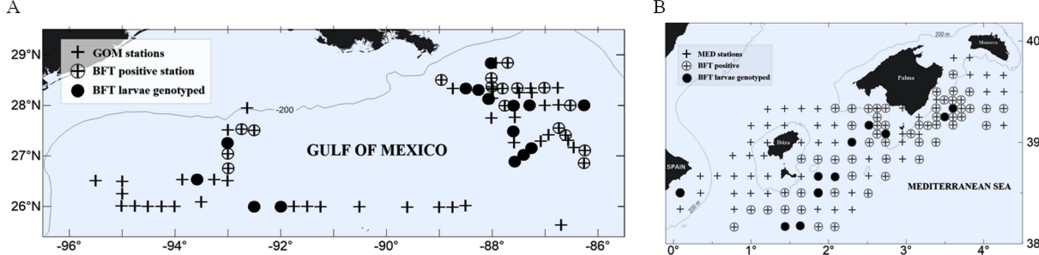

**Figure 1 Study area and larvae collection.** Atlantic bluefin tuna (ABFT) larvae were collected with plankton nets during 2014 in two main spawning areas for *Thunnus thynnus*: (A) the Gulf of Mexico (GOM) and (B) the Mediterranean Sea (MED). Explored stations are indicated with crosses, positive stations for ABFT are indicated with empty circles, and filled circles indicate stations where larvae collected were genotyped. In the GOM larvae were collected in two areas, west and east of 90°W. In the MED larvae were collected in the waters surrounding the Balearic Islands.

## Genetic analysis of ABFT larvae

A total of 112 ABFT larvae spawned contemporarily in the GOM (62 larvae, 30 from eGOM and 32 from wGOM) and MED (50 larvae) were genotyped (Data S1). DNA extraction from ABFT larval tissue was performed with the NucleoSpin® Tissue XS (MACHEREY-NAGEL, Dueren, Germany), and PCR amplification of microsatellite loci alleles was carried out as described in *Uriarte et al. (2019)* with modifications (Table S1). Guts were removed prior to analysis to avoid interference from potential piscivory or cannibalism as observed by *Uriarte et al. (2019)* (except for 18 individuals for which low standard length excluded the possibility of piscivory).

We analyzed eight highly polymorphic microsatellite loci selected according to their moderate-to-high allele count and compatible amplification in two multiplex PCR reactions (Table S1). Allele calling after fragment analysis (STAB VIDA, Caparica, Portugal) was performed with GeneMapper® software v4.0 (Applied Biosystems, Foster City, CA, USA) by two independent readers. We used GeneScan™ 500 LIZ® (Applied Biosystems, Foster City, CA, USA) as size standard. The internal threshold for missing data per individual was 25% (six genotyped loci), which allows for an overall maximum missing data of 13% for locus Tth16-2. A fragment of ~450 bp of the mitochondrial control region (*Viñas & Tudela, 2009*), was sequenced in a subset of genotyped larvae (22 from GOM and 21 from MED). Partial mtDNA control region sequences were assigned to *T. thynnus* based on sequence similarity searches performed with the nucleotide Basic Local Alignment Search Tool (BLAST) (https://blast.ncbi.nlm.nih.gov/). Quality trimmed sequences left a fragment of 361 bp aligned detecting 53 haploid binary sites with SNPs. All sequences have been submitted to GenBank (accession numbers MT912036–MT912078).

## Statistical analysis of ABFT genetic diversity

Examination of microsatellite genetic data was based on allele frequency estimation with GenAlEx software v6.5 (*Peakall & Smouse, 2012*) to assess diversity through observed ($H_O$) and expected ($H_E$) heterozygosity. Software Cervus v3.0.7 (*Kalinowski, Taper &*

*Marshall, 2007*) was used to calculate the polymorphic information content (PIC) and to obtain null allele frequencies using the Maximum Likelihood estimator of *Summers & Amos (1997)*. MICRO-CHECKER (*Van Oosterhout et al., 2004*) was used to further assess null alleles and genotyping errors. GENEPOP 4.7 (*Raymond & Rousset, 1995*; *Rousset, 2008*) was used to calculate the inbreeding coefficient $F_{IS}$ (*Weir & Cockerham, 1984*) and compliance to equilibrium of Hardy–Weinberg (HW). We followed *Weir (1996)* correcting probability values due to low counts for certain genotypes with the Markov Chain Monte Carlo (MCMC) approximation (involving 10,000 dememorization steps, 1,000 batches and 10,000 iterations per batch). The presence of full siblings was screened through replicate sibship analyses with multilocus genotypes excluding individuals with missing data. Analyses were run with COLONY (*Wang, 2004*; *Jones & Wang, 2010*), assuming an inbreeding polygamous model, performing ten very long runs to assess maximum likelihood sibship. Genotype accumulation curves were explored in *R* free software (*R Team, 2018*) in RStudio (*RStudio Team, 2019*) with the "poppr" package (*Kamvar, Tabima & Grünwald, 2014*).

The software GenAlEx v6.5 (*Peakall & Smouse, 2012*) calculated fixation indexes to assess partitioning of genetic diversity at microsatellite loci and at the mtDNA control region within subpopulations relative to the total population. The Wright's F statistic $F_{ST}$, $G''_{ST}$ (*Meirmans & Hedrick's (2011)* standardized $G_{ST}$ further corrected for bias for small $k$ populations), and *Jost's (2008)* D estimate of differentiation were calculated, and the associated probability (p) was obtained based on a 999 data permutation test. For ABFT mtDNA haplotypes the $F_{ST}$ analogous $Phi_{PT}$ was obtained through Analysis of Molecular Variance (AMOVA) of genetic distances. Spatial genetic analysis of both nuclear and mitochondrial diversity was studied performing a Mantel test with GenAlEx software v6.5 (*Peakall & Smouse, 2012*). Mantel correlation coefficient *Rxy* for matrix and spatial autocorrelation within distance classes were performed calculating r by bootstrapping.

Two approaches, Bayesian algorithms and Discriminant Analysis of Principal Components (DAPC), proportionally assigned individuals to population clusters inferred from the microsatellite genetic data. In the Bayesian approach, STRUCTURE v2.3.4 software (*Pritchard, Stephens & Donnelly, 2000*) was used to analyze ABFT larvae genotyped at six microsatellite loci assuming an ancestry admixture correlated allele frequency model (*Falush, Stephens & Pritchard, 2003*), and considering prior sampling location information. Running parameters were set to a burn-in of $2.5 \times 10^4$ followed by a MCMC simulation of $5 \times 10^4$ runs simulating $K$ 1 to 3 populations in 25 iterations. STRUCTURE results were explored with Structure Harvester (*Earl & vonHoldt, 2012*). Clumpak (*Kopelman et al., 2015*) was used to detect the consensus solutions for two $K$ clusters that best fit the Bayesian algorithm used by STRUCTURE. In the DAPC approach we used the R package *adegenet* (*Jombart, 2008*) to study differences among clusters identified from ABFT larvae genotyped at six microsatellite loci, and also from polymorphisms in mtDNA.

## RESULTS

### Field collection and genetic characterization of ABFT larvae

During the 2014 spawning season, approximately half of the stations explored were positive for ABFT larvae (Fig. 1), 43% for GOM and 55% for MED. We genetically characterized a total of 112 ABFT larvae spawned contemporarily in the GOM (62 larvae) and MED (50 larvae), excluding overlap in generations. In the GOM, two sample subsets were separated east (30 larvae) and west (32 larvae) of 90°W according to oceanographic features known to influence ABFT catches (*Teo & Block, 2010*; *Muller-Karger et al., 2015*), with the eGOM dominated by the Loop Current shedding large anti-cyclonic eddies, which generate mesoscale cyclonic and anti-cyclonic eddies that are the key features of the wGOM, preferred by breeding ABFT, and influencing larval fish distribution (*Lindo-Atichati et al., 2012*). Mean SL and standard deviation in mm were 4.94 ± 1.05 (MED) and 4.55 ± 0.64 (GOM) (wGOM 4.49 ± 0.49 and eGOM 4.62 ± 0.77).

The mean number of larvae genotyped at eight microsatellite loci are shown in Table S2, for each of the two spawning areas (GOM and MED), or for the three separate geographical regions (wGOM, eGOM and MED again). The number of individuals was sufficient to adequately quantify allele frequencies (*Hale, Burg & Steeves, 2012*) and further analyze diversity through heterozygosity. The overall number of alleles ranged from six for locus Tth157 to 25 for locus Tth208 (see Table S2), with the highest polymorphic information content (PIC) score (PIC 0.9) obtained for loci Tth208 and Tth1-31. Scoring errors due to stuttering or large allele dropout were excluded with MICRO-CHECKER (*Van Oosterhout et al., 2004*), whereas null alleles may be present at loci Ttho1 and Tth16-2 according to homozygote excess and null allele frequency (Table S2). Deviations from expected proportions in large populations in HW equilibrium imply non-random mating, selection for certain genotypes, mutations, or small population sizes. Under the assumptions of HW equilibrium, the inbreeding coefficient ($F_{IS}$) varies between −1 and +1, measuring the difference between expected heterozygosity ($H_E$) and observed heterozygosity ($H_O$) according to allele frequencies. Significant deviations from HW proportions and homozygote excess (positive $F_{IS}$) were obtained for loci Tth16-2 and Ttho1 (Table 1), as expected from their null allele frequencies (Table S2). Locus Tth16-2 is the microsatellite that deviated most from HW proportions in all populations and areas (Table 1). Close to zero $F_{IS}$ and low null allele frequency did not explain deviation from proportions expected in HW equilibrium for loci Tth157 and Ttho7 in the GOM dataset (Table 1).

Characterization of genetic diversity in larval ABFT may be biased if closely related individuals are collected during plankton sampling. The full-likelihood method implemented in COLONY (*Wang, 2004*; *Jones & Wang, 2010*) can perform sibship analysis to detect full siblings, and is better than other software for parentage analysis in natural populations (*Harrison et al., 2013*). In both spawning areas, sibship was inferred from ~50 replicate multilocus genotype datasets and the resulting parameters averaged (Table S3). The hypothesized numbers of families were high and very similar between

**Table 1 Genetic diversity of Atlantic bluefin tuna (ABFT) larvae genotyped at eight microsatellite loci.**

| Pop[1] | GOM | | | | | | | | | | | | MED | | | |
| | wGOM | | | | eGOM | | | | GOM (wGOM and eGOM) | | | | | | | |
| Locus | $H_O$ | $H_E$ | $F_{IS}$ | HW[2] | $H_O$ | $H_E$ | $F_{IS}$ | HW[2] | $H_O$ | $H_E$ | $F_{IS}$ | HW[2] | $H_O$ | $H_E$ | $F_{IS}$ | HW[2] |
|---|---|---|---|---|---|---|---|---|---|---|---|---|---|---|---|---|
| Tth208 | 0.867 | 0.906 | 0.060 | 0.702 | 0.900 | 0.930 | 0.049 | 0.315 | 0.883 | 0.926 | 0.054 | 0.407 | 0.940 | 0.917 | −0.015 | 0.589 |
| Tth1-31 | 0.938 | 0.877 | −0.053 | 0.932 | 0.867 | 0.883 | 0.035 | 0.776 | 0.903 | 0.886 | −0.012 | 0.982 | 0.840 | 0.871 | 0.046 | 0.402 |
| Ttho7 | 0.815 | 0.837 | 0.046 | 0.239 | 0.821 | 0.845 | 0.046 | 0.157 | 0.818 | 0.844 | 0.039 | 0.006** | 0.820 | 0.836 | 0.030 | 0.707 |
| Tth34 | 0.688 | 0.693 | 0.024 | 0.354 | 0.667 | 0.668 | 0.019 | 0.368 | 0.677 | 0.688 | 0.023 | 0.130 | 0.740 | 0.770 | 0.049 | 0.491 |
| Ttho4 | 0.688 | 0.764 | 0.115 | 0.273 | 0.800 | 0.726 | −0.085 | 0.751 | 0.742 | 0.755 | 0.026 | 0.899 | 0.640 | 0.738 | 0.143 | 0.278 |
| Ttho1 | 0.594 | 0.687 | 0.151 | 0.462 | 0.400 | 0.534 | 0.267 | 0.019* | 0.500 | 0.625 | 0.208 | 0.025* | 0.440 | 0.638 | 0.319 | 0.005** |
| Tth157 | 0.633 | 0.646 | 0.036 | 0.022* | 0.433 | 0.457 | 0.068 | 0.077 | 0.533 | 0.562 | 0.059 | 0.004** | 0.660 | 0.594 | −0.102 | 0.943 |
| Tth16-2 | 0.333 | 0.584 | 0.447 | 0.002** | 0.217 | 0.549 | 0.618 | 0.000*** | 0.277 | 0.577 | 0.529 | 0.000*** | 0.280 | 0.523 | 0.472 | 0.000*** |

Notes:
[1] Pop refers to each area in which ABFT larvae were collected.
[2] Significance after Bonferroni correction (α 0.05) for multiple comparisons at $p < 0.0016$.
* $p < 0.05$.
** $p < 0.01$.
*** $p < 000.1$.
Genetic diversity indicated for each area in which ABFT larvae were collected (Pop), and measured as $H_O$, observed heterozygosity; $H_E$, expected heterozygosity; or $F_{IS}$ inbreeding coefficient calculated according to *Weir & Cockerham (1984)*. HW indicates probability $p$ value obtained with the exact probability test for Hardy–Weinberg equilibrium calculated by the Markov chain method (10,000 dememorization, 1,000 batches, 1,0000 iterations per batch) and level of significance.

spawning areas, $45 \pm 4$ in the GOM and $47 \pm 1$ in the MED, supporting unbiased characterization of genetic diversity.

## Spatial structuring of ABFT genetic diversity

Structuring of genetic diversity between ABFT spawning areas was assessed from multilocus genotypes considering all genotyped microsatellite loci or excluding those with null alleles (Tth16-2 and Ttho1). Partitioning of the total expected genetic diversity between subpopulations or groups under the assumption of equilibrium was quantified through several statistics ranging from 0 to +1. We calculated the general or most frequently used fixation index $F_{ST}$, originally established by Wright for two allele systems, and we also obtained *Meirmans & Hedrick's (2011)* standardized $G''_{ST}$ and *Jost's (2008)* D estimate of differentiation, which are standardized relative to marker heterozygosity and are thus more appropriate estimators for highly polymorphic microsatellite loci. When we only consider genotypes at six microsatellite loci, excluding the two loci with null alleles, genetic differentiation is captured by the value of $G''_{ST}$ and D. Close to zero values with significant associated probabilities for deviation of homogeneity or equilibrium were obtained for GOM versus MED comparisons (Table 2). Genetic diversity partitioning between MED and the two subregions wGOM and eGOM, resulted in higher estimator values and levels of significance for MED versus the wGOM, in support of greater connectivity between ABFT breeding in the MED and in the eGOM during 2014. Pairwise comparisons of wGOM and eGOM genotypes supported genetic homogeneity, with non-significant probabilities associated with close to zero statistics $F_{ST}$, $G''_{ST}$ and D, excluding structuring of genetic diversity within the GOM spawning area.

**Table 2 Pairwise comparison of diversity for indicated spawning areas obtained from ABFT larvae genotypes at indicated number of microsatellite loci.**

| Genotype[1] | $F_{ST}$ | | | $G''_{ST}$ | | | D | | | $p^2$ | | |
|---|---|---|---|---|---|---|---|---|---|---|---|---|
| | 8 loci | 6 loci | 4 loci | 8 loci | 6 loci | 4 loci | 8 loci | 6 loci | 4 loci | 8 loci | 6 loci | 4 loci |
| GOM vs. MED | 0.009 (0.019) | 0.010 (0.001) | 0.011 (0.001) | 0.029 (0.019) | 0.047 (0.001) | 0.070 (0.001) | 0.021 (0.018) | 0.037 (0.001) | 0.059 (0.001) | * | ** | ** |
| wGOM vs. MED | 0.013 (0.012) | 0.012 (0.005) | 0.014 (0.003) | 0.044 (0.014) | 0.055 (0.005) | 0.087 (0.003) | 0.033 (0.012) | 0.045 (0.005) | 0.073 (0.003) | * | ** | ** |
| eGOM vs. MED | 0.010 (0.106) | 0.011 (0.014) | 0.012 (0.019) | 0.020 (0.109) | 0.041 (0.017) | 0.056 (0.019) | 0.015 (0.109) | 0.032 (0.017) | 0.047 (0.019) | n.s. | * | * |
| wGOM vs. eGOM | 0.011 (0.239) | 0.009 (0.404) | 0.009 (0.339) | 0.013 (0.239) | 0.004 (0.401) | 0.008 (0.339) | 0.009 (0.239) | 0.003 (0.401) | 0.006 (0.339) | n.s. | n.s. | n.s. |

Notes:
[1] Data sets including genotypes for eight microsatellite loci (Tth208, Tth1-31, Ttho7, Tth34, Ttho4, Ttho1, Tth157 and Tth16-2), for six loci excluding Tth16-2 and Ttho1 due to null alleles, and at four loci conserving microsatellites in HW equilibrium with higher PIC (Tth208, Tth1-31, Tth34 and Ttho4).

[2] Significance after Bonferroni correction (α 0.05) for multiple comparisons at $p < 0.0042$ is indicated underlined.

\* $p < 0.05$.

\*\* $p < 0.01$.

Genetic diversity estimators $F_{ST}$, $G''_{ST}$ (Hedrick's standardized $G_{ST}$ further corrected for bias for small $k$ populations) and Jost's D estimate of differentiation. Associated probability (p) indicated in brackets obtained through 999 data permutations and summarized in the last column as non-significant (n.s.) or according to significance levels of $p < 0.05$ (\*) and $p < 0.01$ (\*\*).

Unique individuals were completely discriminated with four loci according to a genotype accumulation curve (Fig. S1). We obtained diversity estimators from ABFT larvae genotypes at the four most informative loci (PIC ≥ 0.7) that were neutral or in accordance with HW proportions, thus informing of homogeneity of diversity between larval groups. Table 2 shows that four neutral microsatellite loci: Tth208, Tth1-31, Ttho4 and Tth34; detected structuring between larvae spawned in breeding grounds of each ABFT stock, underlining the power of resolution of microsatellite loci as markers for this tuna species. Even multilocus genotypes for only the two loci with the highest PIC were able to significantly detect structuring between GOM and MED larvae (data not shown).

Pairwise comparison of MED and GOM larvae haplotypes at 53 mtDNA control region variable sites (SNPs) through AMOVA resulted in a PhiPT value of 0.029 (with a significant associated data randomization probability of 0.019 with 999 permutations). Phylogenetic analysis of mtDNA sequences to infer evolutionary relationships from a common ancestor failed to separate larvae spawned in the MED and in the GOM (data not shown), as previously reported (*Alvarado Bremer et al., 2005*; *Carlsson et al., 2007*).

ABFT genetic diversity partitioning, assessed herein exclusively through larvae resulted in fixation indexes within the ranges reported thus far, analyzing polymorphism in ABFT mtDNA sequences, microsatellite loci or SNPs, in samples of certain (larvae) or almost certain (juveniles) origin (*Carlsson et al., 2007*; *Puncher et al., 2018*; *Rodríguez-Ezpeleta et al., 2019*) (Table 3). We collected ABFT larvae in a precise spatial and temporal frame, and our results are consistent with other genetically characterized juvenile ABFT and larvae collected in different years and areas compiled in Table 3. Besides $F_{ST}$, we report other genetic diversity estimators, $G''_{ST}$ (*Meirmans & Hedrick's (2011)* standardized $G_{ST}$ further corrected for bias for small $k$ populations) and D (*Jost's et al. (2018)* estimate of differentiation), as they show a higher variance, and are complementary measures of

**Table 3 Summary of studies assessing ABFT genetic structuring between main spawning areas (GOM and MED) through individuals of early life stages (larvae or young-of-the-year YOY) ensuring correct management unit assignment.**

| Reference | Genetic tool | ABFT stock[4] | | Fixation index[5] |
|---|---|---|---|---|
| | | **Western Atlantic** | **Eastern Atlantic** | |
| (Carlsson et al., 2007) | mtDNA sequence[1] | Gulf of Mexico (GOM) Larvae (40) Year 2003 | Mediterranean Sea (MED) YOY (107) Years 1998, 1999, 2000, 2001, 2002. | 0.013* |
| | Microsatellite loci[2] | GOM Larvae (40) Year 2003 | MED YOY (280) Years 1998, 1999, 2000, 2001, 2002. | 0.006*** (8 loci) |
| (Puncher et al., 2018) | SNP panel[3] | GOM (64) and Cape Hatteras (16) Larvae (64) and YOY (16) Years 2007, 2008, 2009, 2010. | MED Larvae (63) and YOY (350) Years 2008, 2011, 2012, 2013. | 0.008* (95 SNP panel) 0.014* (58 SNP panel) 0.034*** (24 SNP panel) |
| (Rodríguez-Ezpeleta et al., 2019) | SNP panel[3] | GOM (26) and Slope Sea (13) Larvae (26) and YOY (13) Years 2007, 2008, 2009, 2010. | MED Larvae (48) and YOY (117) 2008, 2011, 2012, 2013. | 0.004 (n.i.) |
| This study | mtDNA sequence[1] | GOM Larvae (22) Year 2014 | MED Larvae (21) Year 2014 | 0.029* |
| | Microsatellite loci[2] | GOM Larvae (62) Year 2014 | MED Larvae (50) Year 2014 | 0.009* (8 loci) 0.010** (6 loci) 0.011** (4 loci) |

**Notes:**
[1] The control region was sequenced in 847 bp (Carlsson et al., 2007) or 361 bp (this study).
[2] Loci Tth5, Tth8, Tth10, Tth21, Tth34, Ttho-1, Ttho-4, and Ttho-7 were genotyped in Carlsson et al. (2007), and for this study we analyzed loci Tth208, Tth1-31, Ttho7, Tth34, Ttho4, Ttho1, Tth157, and Tth16-2.
[3] SNPs derived from genomewide search for spatially informative loci by restriction site-associated DNA sequencing (RAD-seq).
[4] Phase of early life stage (number of individuals) and collection year.
[5] Significant probability obtained by 999 random permutations (*$p < 0.05$, **$p < 0.01$ and ***$p < 000.1$), n.i. not indicated. For nuclear markers $F_{ST}$ is indicated. For mtDNA $Phi_{ST}$ was calculated in Carlsson et al. (2007), we calculated analogous $Phi_{PT}$ for this study.
* $p < 0.05$.
** $p < 0.01$.
*** $p < 000.1$.
For each study the genetic tool and features of the collection of ABFT individuals used to characterize genetic diversity are summarized. The reported fixation indexes from pairwise comparison between ABFT collected in Western or Eastern stocks is compiled in the last column.

fixation and allelic differentiation (Bird et al., 2011), arguing against the generalized exclusive use of $F_{ST}$.

We assessed whether genotypes were more similar when comparing larvae collected in the same spawning area than when comparing larvae from different spawning areas. Spatial genetic analyses were performed by analyzing the autocorrelation between geographic and genetic data through a Mantel test. Close to zero autocorrelation statistics resulted from analyzing genotypes for six loci (R 0.127, $p$ 0.001) and mtDNA control region binary haplotypes (R 0.051, $p$ 0.022). Genetic structuring within different distance classes (within the same or different spawning areas) was explored through multivariate spatial autocorrelation analysis. Autocorrelation for microsatellite genotypes resulted in a positive r value of 0.024 (with confidence limits of 0.005 and −0.004) for distance classes within the same spawning area and a negative r value of −0.020 (with U of 0.003 and L of −0.004) for distance classes belonging to different spawning areas, in agreement with a weak but significant spatial structuring of genetic diversity.

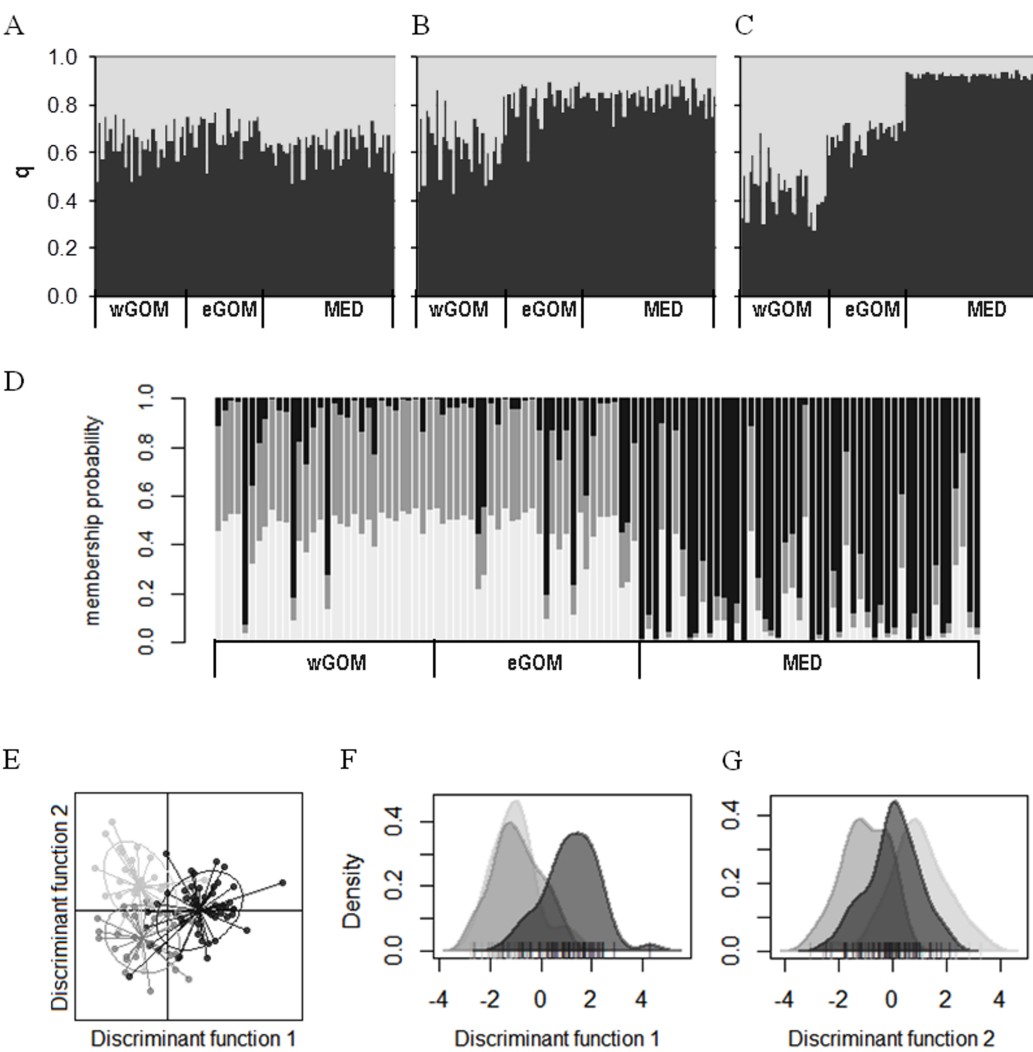

**Figure 2 Clustering of ABFT larvae genetic diversity.** Genotypes at six microsatellite loci (Tth208, Tth1-31, Ttho7, Tth34, Ttho4 and Tth157) were used to characterize genetic diversity of ABFT larvae spawned in the MED (black), and GOM (grey). GOM can be segregated into wGOM (light grey) and eGOM (dark grey). (A–C) Bayesian clustering of ABFT larvae performed with *STRUCTURE v2.3.4* (*Pritchard, Stephens & Donnelly, 2000*) software through admixture modeling considering prior information on ancestry from the collection area. For each larva the proportion of ancestry (q) for each of two population clusters is plotted considering prior ancestry information fitting 68% (A), 20% (B), or 12% of the data (C). (D) Discriminant Analysis of Principal Components (DAPC) performed with R package *adegenet* to show probability membership to three clusters through one discriminant function with an eigen value of 130.6 (40 principal components accumulating 0.925 variance). (E–G) Two discriminant functions obtained from DAPC (with eigen values of 64.72 and 29.34). Dot plot (E) and density plots (F, G) for each function are represented to illustrate overlapping of MED and GOM genetic diversity.

We investigated connectivity and structuring of genetic diversity through assignment of individual ABFT larvae to two *K* clusters through a Bayesian analysis of multilocus genotypes in an admixture model considering prior location information (Figs. 2A–2C). In two out of three consensus solutions, representing 32% of all simulations, a gradient from wGOM to MED was observed in the proportion of assignment of each larva to two

clusters, with eGOM appearing as an intermediate or mixing area (Figs. 2B and 2C). Figure S2 shows the statistic ΔK (deltaK) proposed by *Evanno, Regnaut & Goudet (2005)* to detect the real number of clusters based on the rate of change of probabilities between successive K values, and that for our data is necessarily only obtained for K = 2. A different clustering approach is multivariate DAPC analysis that is applied to discriminate clusters through discriminant analysis performed on data previously transformed through principal components analysis (*Jombart, Devillard & Balloux, 2010*). Assignment of individual ABFT larvae to clusters performing DAPC with one discriminant function (DF) clearly separated larvae spawned in the GOM and in the MED (Fig. 2D). DAPC analysis of variable sites found in the mtDNA control region sequences also separated larvae spawned at Eastern and Western stock reproduction areas (Fig. S3). In Fig. 2D, the similar proportion of assigned individuals to the wGOM and the eGOM supports absence of structuring within the GOM spawning area. Retaining 0.925 variance in two DFs separated ABFT spawned in wGOM, eGOM and MED in three overlapping clusters (Fig. 2E). DAPC is thus in accordance with the wGOM to MED gradient observed through Bayesian clustering, and supports apparent connectivity between the ABFT Western and Eastern stocks. Separate density plots (Figs. 2F and 2G) for DF1 (eigen value 64.72) and DF2 (eigen value 29.34) also illustrate homogeneity between larvae spawned in the GOM.

## DISCUSSION

This initial study analyzes ABFT genetic structure exclusively through larvae spawned contemporarily in the GOM and in the MED, and to our knowledge structuring within the ABFT GOM spawning grounds is hereby investigated for the first time. In accordance to previous studies summarized in Table 3, we conclude that there is weak genetic structuring between ABFT stocks through nuclear microsatellites and mitochondrial sequences, which excludes philopatric dispersal of ABFT, as no differences were found between markers inherited biparentally (microsatellites) or through the breeding females (mtDNA). In contrast, in other highly migratory marine species, sex-biased gene flow does result in complex population structure. This is the case of the loggerhead turtle, a species in which population structuring is detected to increase with life stages according to maternally inherited mtDNA, as opposed to absence of differentiation in microsatellite loci due to the constant gene flow driven by the males during migration (*Bowen et al., 2005*). For the case of highly migratory tuna populations with high levels of gene flow, this study supports the use of larvae and microsatellites for population genetics, detecting weak structuring between larvae spawned in the main breeding grounds of large sized and highly mixed populations with only four neutral loci. The use of microsatellites in population genetics has diminished due to the emergence of high throughput sequencing technologies (*Ovenden et al., 2015*; *Cuéllar-Pinzón et al., 2016*), with certain studies reporting that 4–12 SNPs are equivalent to one microsatellite locus (*Guichoux et al., 2011*). Two studies have analyzed SNPs in ABFT collections detecting weak structuring between management units with as few as 36 SNPs (*Rodríguez-Ezpeleta et al., 2019*) or 24 SNPs (*Puncher et al., 2018*), with $F_{ST}$ obtained from GOM-MED comparisons ranging from 0.004 to 0.034, respectively. *Antoniou et al. (2017)* characterized ABFT adult samples
collected in the MED through genome-wide SNPs and 16 microsatellite loci, but only found statistically significant close to zero $F_{ST}$ values for microsatellite multilocus genotypes. Studies addressing identification of microsatellites in ABFT are scarce with some of the loci selected for the present study assessed in different ABFT collections (*Carlsson et al., 2004*, *2007*; *Riccioni et al., 2010*, *2013*; *Vella et al., 2016*; *Antoniou et al., 2017*). For future monitoring of ABFT population dynamics genetic tools ought to be standardized.

The use of early life stages for population genetics in highly migratory species such as ABFT, that lack barriers to gene flow, is important as it provides the genetic signal from successful breeders. Sibship analysis did not reveal impartiality in sampling related individuals when collecting larvae in GOM or MED spawning areas, which is consistent with previous studies reporting highly unrelated collections of ABFT larvae and juveniles (*Puncher et al., 2018*). The larvae analyzed in this study belonged to a single generation and were theoretically drawn from single and randomly mating units with cluster analysis supporting detectable connectivity between eGOM and MED spawning areas. Our separation of wGOM and eGOM data to assess HW equilibrium was artificial and no structuring between GOM subregions was found. However, proportions expected in HW for loci Ttho7 and Tth157, with increased significant deviation for pooled data, could be interpreted as a recent sign of non-random mating between reproducing adults within the GOM. A hypothetical resident population of breeding ABFT in the wGOM would agree with increased gene flow between stocks in the eGOM, as suggested by pairwise genetic diversity estimators corrected for multiple comparisons. Bearing in mind we did not find structuring within the GOM, the fact that clustering presents the eGOM as an intermediate overlapping area between MED and wGOM also supports the existence of mixing areas for breeders from each stock, which we propose would occur in the eGOM. A scenario with ABFT adults migrating from the MED or the eastern northern Atlantic Ocean to spawn in the eGOM would be plausible considering their larger population size and the observed spatial patterns demonstrating transoceanic migrations, in addition to the similar trophic baselines reported for the eGOM and MED environments (*Laiz-Carrión et al., 2015*).

Conservation and management of ABFT demand continuous research into connectivity between Western and Eastern ABFT stocks, with variable rates of movement and population exchange in mixing hotspots in the North Atlantic Ocean (*Rooker et al., 2019*), and discrepancy between environmental and genetic profiles reported recently for some adults (*Brophy et al., 2020*). Several studies have in fact shown there is more movement from the Eastern to the Western stock than vice versa (*Block et al., 2005*; *Rooker et al., 2008b*; *Taylor et al., 2011*), and reproductive mixing between stocks is proposed to occur in the Slope Sea (*Richardson et al., 2016*). *Puncher et al. (2018)* and *Rodríguez-Ezpeleta et al. (2019)* evaluated the assignment power of their respective traceability SNP panels that were specifically selected based on their capacity to differentiate stocks analyzing YOY and larvae (in lower or similar numbers to this study, see Table 3). In agreement with our results indicating connectivity between eGOM and MED, they found incorrect assignment for certain individuals of known natal origin. *Puncher et al. (2018)* found the proportion of

larvae and YOY juveniles poorly assigned was higher in the western Atlantic Ocean compared to the MED. *Rodríguez-Ezpeleta et al. (2019)* found higher percentage of incorrect assignment for GOM (10%) than for MED (2%); and more intriguingly, assigned larvae from the Slope Sea to both management areas, which could be in agreement with interbreeding of GOM and MED tuna in that region, and according to our results this hypothetical scenario would extend to the eGOM. In summary, genetic tools are able to detect weak ABFT population structuring by analyzing biological material obtained from larval stages, indicating apparent connectivity between GOM and MED spawning grounds, and calling for future research into areas favorable for mixing of breeders of separate management units to ensure conservation of genetic diversity.

## CONCLUSIONS

A cooperative effort allowed collection of tuna larvae during the 2014 reproductive season in the GOM and the MED, the main spawning areas for the Western and the Eastern ABFT stocks correspondingly. Genetic diversity was characterized for the first time exclusively from larval individuals of known origin that provide the signal of successful breeders. Significant genetic structuring between larvae spawned in each spawning ground was found, in agreement with previous studies. Segregation of ABFT larval individuals within the GOM according to oceanographically distinct features in eGOM and wGOM does not indicate structuring. Fixation indices and clustering analysis indicated weak but detectable connectivity between ABFT that breed in the MED and in the eGOM was stronger than between adults spawning in the MED and the wGOM, calling for future research into areas favorable for mixing of breeders belonging to different stocks.

## ACKNOWLEDGEMENTS

We appreciate the sampling efforts conducted at sea on R/V F.G. Walton Smith and SOCIB. We are grateful for larval collection, processing and analyses to the laboratories of the NOAA Southeast Fisheries Science Center and the Spanish Institute of Oceanography (Instituto Español de Oceanografía), we thank Amaya Uriarte for sample processing. Valuable comments were provided by Sarah Privoznik. The scientific results and conclusions, as well as any views or opinions expressed herein, are those of the author(s) and do not necessarily reflect those of NOAA or the Department of Commerce.

### Funding

This collaborative study was supported by "ECOLATUN" PROJECT CTM2015-68473-R (MINECO/FEDER) funded by Spanish Ministry of Economy and Competitiveness; "TUNAGEN" project funded by Instituto Español de Oceanografía (IEO); and "BLUEFIN" project financed by IEO and Balearic Island Observing and Forecasting System (SOCIB). This research was funded by NASA (NNX11AP76G S07), the NOAA National Marine Fisheries Science Service through the Southeast Fisheries Science Center,

as well as by the Cooperative Institute for Marine and Atmospheric Studies under Cooperative Agreement NA15OAR43200064 at the University of Miami, Miami, FL, USA. There was no additional external funding received for this study. The funders had no role in study design, data collection and analysis, decision to publish, or preparation of the manuscript.

### Grant Disclosures

The following grant information was disclosed by the authors:
Spanish Ministry of Economy and Competitiveness: CTM2015-68473-R.
IEO "TUNAGEN".
IEO and Balearic Island Observing and Forecasting System (SOCIB) "BLUEFIN".
NASA: NNX11AP76G S07.
Southeast Fisheries Science Center.
University of Miami: NA15OAR43200064.

### Competing Interests

The authors declare that they have no competing interests.

### Author Contributions

- Carolina Johnstone conceived and designed the experiments, performed the experiments, analyzed the data, prepared figures and/or tables, authored or reviewed drafts of the paper, and approved the final draft.
- Montse Pérez conceived and designed the experiments, analyzed the data, authored or reviewed drafts of the paper, and approved the final draft.
- Estrella Malca conceived and designed the experiments, performed the experiments, analyzed the data, prepared figures and/or tables, authored or reviewed drafts of the paper, and approved the final draft.
- José María Quintanilla analyzed the data, authored or reviewed drafts of the paper, and approved the final draft.
- Trika Gerard conceived and designed the experiments, authored or reviewed drafts of the paper, and approved the final draft.
- Diego Lozano-Peral conceived and designed the experiments, performed the experiments, prepared figures and/or tables, authored or reviewed drafts of the paper, and approved the final draft.
- Francisco Alemany conceived and designed the experiments, performed the experiments, authored or reviewed drafts of the paper, and approved the final draft.
- John Lamkin conceived and designed the experiments, authored or reviewed drafts of the paper, and approved the final draft.
- Alberto García conceived and designed the experiments, authored or reviewed drafts of the paper, and approved the final draft.
- Raúl Laiz-Carrión conceived and designed the experiments, analyzed the data, authored or reviewed drafts of the paper, and approved the final draft.

## Field Study Permissions

The following information was supplied relating to field study approvals (i.e., approving body and any reference numbers):

A total of 76 stations were sampled in the GOM cruise "WS1405" (approved by the National Oceanic and Atmospheric Administration with Permit Number TUNA-SRP-14-02) carried out from 28 April to 20 May 2014 aboard the R/V F.G. Walton Smith.

The "BLUEFIN14" cruise (approved by the Spanish Institute of Oceanography (Instituto Español de Oceanografía) "BLUEFIN TUNA" project) explored a total of 123 stations and took place from 13 June to 3 July 2014 in the western MED on board the R/V SOCIB, of the Balearic Islands Coastal Observing and Forecasting System.

## Data Availability

The sequences are available at GenBank: MT912036–MT912078.

## Supplemental Information

Supplemental information for this article can be found online at http://dx.doi.org/10.7717/peerj.11568#supplemental-information.

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
