# Peer review of "Genetic connectivity between Atlantic bluefin tuna larvae spawned in the Gulf of Mexico and in the Mediterranean Sea"

_PeerJ, doi:10.7717/peerj.11568_

## Round 0.1 · original submission · Major Revisions

As the reviewers mentioned that the analysis of larvae is a strong point of the study but the rest of the work has already been extensively published; I would like to request the authors to make major revision focusing and highlighting their work on larvae.

Reviewer 1 ·

Basic reporting

The present study was same to those of previous studies.

Experimental design

There are a few studies to assess using "larvae" but fundamental methodology is similar to previous studies.

Validity of the findings

Same and similar results have been found in proviso studies.

Additional comments

The current study tried to understand stock discrimination between Atlantic bluefin tuna larvae by means of molecular genetic analyses. As authors stated and cited, there are a number of similar research have been published, although there is no study have been examined in larvae. However, there are no difference between present and previous studies, and I do not see any novelty except for authors used larval samples. I know larval collection is not easy as it needs to allocate plenty of time and expenditure for research cruise. However, stock analysis is not necessary to use larval samples only as previous studies used juvenile and adult samples. I suppose authors need to collect larvae during spawning period to assess temporal variation because spatial assessment has been done by previous studies. I feel the present study does not overwhelm previous same, similar and relevant studies by means of molecular, otolith and behavioural analyses in the Atlantic bluefin tuna. Although authors stated “Larval early life stages have scarcely been studied to decipher ABFT population structure despite providing the genetic signal from successful breeders” in the first sentence of Discussion, stock discrimination and connectivity research are not necessary examined for larval samples. Furthermore, the current discussion is redundant, e.g., L307—L323 should be in Introduction and/or Results and/or Materials methods. I do not understand why authors still debate about molecular methodology in L324—L341 because there have been done similar or same research in the species and other fish species. Such the discussion seems to be no longer interesting. Authors should focus on population structure, divergence and connectivity only together with conservation and management of the species (L342 onwards are useful discussion). Overall, I cannot recommend the paper is published in Peer J because of lacking of novelty.

Reviewer 2 ·

Basic reporting

Language is appropriate, clear and unambiguous. Introduction is partially to be revised as stated in the comments to authors. Literature references are adequate. Iconography and structure of article is adequate.

Experimental design

Adequate and sound. Hypothesis to be tested and aims are clearly stated. The described research fits the aims and scope and standard of the journal.

Validity of the findings

No comment. Please refer to the Comments to Authors.

Additional comments

Johnstone et al. carried out a population genetic study on Atlantic bluefin tuna (ABFT) larvae using microsatellite and CR sequence markers, providing a substantial corroboration of already known ecological and population dynamic features of ABFT but also providing new evidence on the occurrence of a panmictic or near panmictic population in the Gulf of Mexico (the well-known western spawning area) and, with some but not all the statistical tests, a possible, very weak but detectable, reproductive connectivity between the eastern Mediterranean population and ABFT inhabiting the eastern part of the Gulf. The strengthen of this work is the unprecedented exclusive use of ABFT larvae that were collected according a very solid and informative sampling design (all main ABFT spawning areas covered within the same spawning season and year). Results are largely consistent in terms of levels of genetic differentiation and of significance patterns and authors appropriately the use of a limited number of neutral microsatellite loci that have been repeatedly proven and used for polymorphisms and replicability by numerous previous research works. However, the use of a mtDNA partial sequence marker that is highly-variable but less informative for population genetic purposes appears to weak the analytical power of their analyses and issue from phylogeographical analyses are effectively redundant with respect to those already published by several other authors. Results were adequately and deeply discussed highlighting advances and confirmation of already known ABFT population genetics’ patterns.
Therefore, the research paper contributes to shape ABFT population dynamics and connectivity providing some foreseen regarding additional features to those already scientifically consolidated.


Major and Minor comments
Abstract
l. 31 “supporting connectivity between larvae”: larvae cannot reproductively interact. Spawning populations or mature individuals can interact reproductively. Rephrase the sentence more appropriately and with correct meaning.
ll. 38-39, “leading to a strong connectivity between populations”. Statistical data and results obtained did not speak in favor of a strong connectivity but rather of a weak but detectable connectivity. Also, not all statistical results accounted for this issue and this should be mentioned in the abstract.
l. 40. Change “management units” to “populations”

Introduction
ll. 61-72. Several population genetic studies carried out along with 15 years have accounted for the genetic and reproductive separation of ABFT into two populations spawning predominantly in the Mediterranean and Gulf of Mexico. Such genetic issues supported those retrieved from a huge number of observations obtained with multidisciplinary approaches (i.e. reproductive biology, ecology, standard, satellite and microchemical tagging that have been integrated since 2011-12 by ICCAT-GBYP program and surveys). Even if the levels of genetic differentiation assessed between population samples collected from the two spawning populations were very low (from ~1‰ to ~1%), they resulted mostly significant and the structure appeared and was interpreted as robust (see all the works from BBlock group and those produced within the ICCAT-GBYP framework). Such low levels of differentiation were interpreted as due to the relative recent origin of the Atlantic ABFT populations (see Alvarado-Bremer et al. 2005) and to the very large population size that strongly reduces genetic drift detectable by neutral markers. I recommend to authors to summarize better and more consistent with results already obtained the ecological-biological and genetic state-of-the-art of ABFT. For example, Block et al. 2005 and Rooker et al. 2008 revealed the intermingling of individuals from the two spawning populations in the Atlantic feeding grounds (e.g. Mid Atlantic Bight, Bay of Biscay, Coast of Maine etc) but also demonstrated that the any ABFT born in the Mediterranean visited the Gulf and viceversa. This should be accounted to provide a scenario more coherent to existing and published data.
ll. 85-87. This sentence is correct but I recommend to authors to acknowledge here that at least four popgen surveys were based on early-life stages (larvae and YOY): Carlsson et al., 2004, 2007; Puncher et al., 2018; Rodriguez-Ezpeleta et al., 2019, almost possessing the same robustness given by the exclusive use of larvae.
ll. 106-108. This info is irrelevant to the purpose of the paper.
ll. 109-111: same criticism pointed for ll. 38-39: change strong with a more prudent and soft adjective. Let be more cautious in postulating that data “support the hypothesis of mixing of breeders belonging to different stocks”. “Could support” or “seem to support” could be better for consistency with the results. In addition, “Statistical analysis” change in “Individual-based clustering analyses”.

Materials and Methods
ll. 135-136. Instead of sentencing “A minimum number of ….” provide here and not in the Results the number of larvae analysed” and coherently provide in the previous paragraph the number of larvae collected.
ll. 153-154. Authors gave in the manuscript the GenBank accession numbers of the deposited mtDNA sequences; however deposited sequences are not yet accessible. On the contrary, any statement on microsatellite data deposition was given by authors. Authors must provide availability of these data.

Results
ll. 266-268. This sentence is strongly consistent with the results obtained and represents most of the statistical issues obtained and is strongly discordant with the sentences in the abstract and at the end of the Introduction. I recommend to homogenize the issues provided along with the ms. At any rate, the paragraph (from ll. 264-272) should be moved in the Discussion.
ll. 301-305: I disagree and these results can be also explained by an incomplete sorting of the gene pool due to the very low genetic drift affecting these large sized populations.
ll. 356-358. This is incorrect. MED is not intermediate. The intermediate genetic cluster refers to eGOM.
l. 382 “GOM” changes into “eGOM”
l. 396 “Fixation indices and clustering analysis indicated connectivity between ABFT that breed in the MED and in the eGOM was stronger… “ changes into “Fixation indices and clustering analysis indicated weak but detectable connectivity between ABFT that breed in the MED and in the eGOM was stronger…“

Reviewer 3 ·

Basic reporting

The manuscript of Johnstone et al is a new study on the population genetic connectivity of ABFT between Mediterranean and Atlantic. The study is well defined with a remarkable extensive effort in the sampling of larvae.
The paper is well structured, well written with affectable structure. No major flaws were detected in the structure, English, Figures and Tables

Experimental design

Several flaws were detected in the design and analyses of the manuscript.
I cannot find the rationale of comparing two submaples within the Gulf of Mexico. All the literature accepts the GoM as a single reproduction are, thus a better description why the authors decided to compare the two samples of GoM is needed.
Accordingly, the authors should better define the objective of the study. Similarly, the author should better describe the sampling procedure. It is really confusing. The section of “Genetic Analysis of ABFT larvae” should be improved. It is difficult to find how many larvae have been analyzed in each location. For instance, the number of specimens analyzed is described in the results section, but it should be also included in material and methods (a summary of the sampling would be appreciated).
The analysis proposed is correct, although the interpretation of some results is confusing. In Table 2, no correction of multiple tests is done, with a possible invalidation of the results and conclusion drawn from the results.

Validity of the findings

The result obtained here are in concordance with the current acknowledge of the population structure of ABFT, which has been extensively published in the literature. Some flaws in the analyses are detected (see previous section) and thus with the possibility of invalidating the results and conclusions.
In addition, the DAPC analysis depicts a weak differentiation between Med and GoM, which is contradictory with the title of the manuscript.

Additional comments

Other changes should be done along the manuscript
Line 45. “They are managed by regional fisheries organizations as four stocks”. This sentence is really confusing. The authors confused stocks and species.
Line 266. There is a conclusion in the results section. Conclusions should be detailed in the discussion and conclusion section.
Table 3. I cannot find the Johnstone et al., 2019 study. This refence is not necessary. Keep only “this Study”.

---

## Round 0.2 · Minor Revisions

After critical evaluation of revised manuscript and rebuttal letter, I am convinced, but you have to address few more points!
Section Editor Dr James Reimer has raised the following valid questions, please address these issues and resubmit-

"After looking over the paper and the latest version, I agree with the AE that the paper has been well revised, and the authors should be commended for that.

However, there are some small issues remaining, such as simple language issues "Partition[ing] of genetic diversity", to a lack of any reference to complex population structure that has been shown to explain similar differences between patterns of genetic structure among juvenile and adult migratory species such as sea turtles; some reference to this issue is needed.

The connectivity discussion needs some work too, such as when the authors say "differences and similarities are better captured by the value of Hedrick´s standardized GST´´ and Jost´s D estimate of differentiation" relative to Fst. All of these metrics measure the same thing and it is simply whether or not the differentiation metric is standardized (G''st and D) or not (Fst) relative to marker heterozygosity that helps people to understand how to interpret the differences between the values (e.g. see Bird et al. 2011).

Thus, based on these issues, I think the paper would be well served by having a thorough going over once more by the authors, particularly with regards to the points above, but not limited to them."

Reviewer 2 ·

Basic reporting

Comments are summarized in the General Comments for the author

Experimental design

Comments are summarized in the General Comments for the author

Validity of the findings

Comments are summarized in the General Comments for the author

Additional comments

Authors addressed all requests and comments I posed to the submitted version of the manuscript, adequately motivating and detailing changes introduced.

---

## Round 0.3 · accepted · Accept

Thank you for your effort to shape the manuscript to be accepted.
Congratulations! Hope you will continue to do research on ABFT and consider PeerJ again for your work to be published.